# Neuronal Mechanisms of Reading Informational Texts in People with Different Levels of Mental Resilience

**DOI:** 10.3390/brainsci14090944

**Published:** 2024-09-21

**Authors:** Małgorzata Chojak, Anna Gawron, Marta Czechowska-Bieluga, Andrzej Różański, Ewa Sarzyńska-Mazurek, Anna Stachyra-Sokulska

**Affiliations:** 1Instytut of Pedagogy, Uniwersity of Marie Curie-Sklodowska, 20-612 Lublin, Poland; anna.gawron@mail.umcs.pl (A.G.); marta.czechowska-bieluga@mail.umcs.pl (M.C.-B.); andrzej.rozanski@mail.umcs.pl (A.R.); ewa.sarzynska-mazurek@mail.umcs.pl (E.S.-M.); 2Instytut of Psychology, Uniwersity of Marie Curie-Sklodowska, 20-612 Lublin, Poland; anna.stachyra-sokulska@mail.umcs.pl

**Keywords:** reading comprehension, cognitive resilience, NIRS, technology, adults, brain

## Abstract

The aim of this study was to verify whether the level of mental resilience would differentiate reading comprehension performance when using different information carriers. More than 150 people filled out a test regarding the level of resilience. They then participated in a survey using fNIRS. Their task was to read a one-page informational text and answer several questions. The results showed no differences in correct answers between groups of people with different levels of resilience. In the groups of people with high and low levels of resilience, the number of correct answers was not differentiated by the type of carrier. Among those with moderate levels of resilience, better results were obtained by those who read text printed on paper. Analyses of neuronal mechanisms showed that the type of carrier differentiated brain activity in each group. Obtaining the same number of correct answers in the test was the result of different neuronal mechanisms activated in those who used a computer and those who read a printed text.

## 1. Introduction

Ego resilience is the continuously developed ability to adequately cope with everyday tasks, including under stressful and changing conditions. A resilient person adequately, consistently and persistently adapts to the environment through the appropriate use of their own abilities as well as environmental factors. 

Mental resilience can be seen as a self-regulatory mechanism involving cognitive, emotional and behavioral elements [1], but also as a broader construct, which plays an imperative role over other individual resources [2]. It includes emotional elements (striving for positive emotions, positive affect and emotional stability), cognitive elements (e.g., beliefs and expectations regarding, among others, perceiving reality as a series of challenges and awareness of one’s own competences) and behavioral elements (taking action in new situations, seeking new experiences, applying diverse and effective coping strategies) [3,4].

Resilience helps us cope with stressful situations [5]. Individuals with higher resilience levels are more likely to perceive stressful situations as a challenge and have positive emotions associated with them, compared to individuals with lower levels of the trait [6]. Resilience helps find solutions and evaluate their usefulness, and boosts confidence. Individuals with low resilience act in a conservative and rigid manner in the face of stress, and their reactions are characterized by chaotic and dispersed behavior [7].

Psychological resilience is linked to cognitive resilience, i.e., the level of cognitive skills that an individual possesses independently or in spite of internal and external conditions [8]. This type of resilience includes the ability to understand written text. Research indicates a correlation between mental resilience and reading competence [9]. Researchers have even defined the term resilience of reading [10]. Among the determinants of this phenomenon are genetic, environmental and educational factors, but also the type of information carrier. Research to date has not conclusively demonstrated whether, among adults, more effective reading is connected to data presented on paper or on a tablet or other electronic medium [11,12,13,14]. In an age of very rapid technological development, such knowledge could be important not only for employers, but also for those using data for therapeutic or educational purposes. It seems important also because the mental and physical burdens associated with the COVID-19 epidemic will continue to be felt in the coming years.

In neuroscience publications, resilience is defined not only as the absence of psychopathology despite exposure to adversity, but also as the ability to function competently in different spheres, despite, for example, a diagnosis of mental disorder [15]. In neurological terms, we can therefore talk about coping with brain pathology and avoiding such pathology [16]. Among the determinants of resilience, in addition to genetic factors [17] or personality traits [18], an important role is played by factors related to brain structure and function [15,19].

Intensive development of brain neuroimaging techniques has facilitated the search for neurobiological determinants of mental and cognitive resilience in studies assessing potential neurobiological correlates of stress susceptibility and/or resilience. To date, most research has focused on:-studies of people with disorders such as post-traumatic stress disorder (PTSD), anxiety, depression or dyslexia due to the correlations found with selected scales for the study of resilience [20];-structural and functional analysis of the brain using EEG and fMRI [21].

The neuroimaging findings indicate that several brain structures are vital here. These include the hippocampus, amygdala, insula, anterior cingulate cortex and prefrontal cortex, as well as related brain networks such as the default mode network and central executive network [22]. Research to date has shown that:-childhood trauma results in negative connectivity between the prefrontal cortex and the amygdala, which may be a predictor of negative anxiety control [23] and difficulties with planning and working memory [24] in adults;-there is a significant positive correlation between an individual’s level of resilience and the thickness of the cortex in the right hemisphere (comprising the lateral occipital cortex, the fusiform gyrus, the inferior parietal cortex, as well as the medial and inferior temporal cortex [25,26]), i.e., reduced resilience is associated with reduced cortical thickness in the areas that are involved in processing of emotional visual stimuli [25];-in children who have experienced abuse and neglect, a reduction in the size of the hippocampus is evident, which also persists into adulthood [27] and results in difficulties related to broadly defined learning process [28];-adaptation of neural network efficacy and engagement of additional brain areas may provide mechanisms for coping with increased pathological load, both socio-emotional and cognitive [29,30,31,32], which is consistent with the cognitive reserve hypothesis, which assumes that cognitive function is preserved through functional adaptations of large-scale networks.

The hippocampus and prefrontal cortex play a leading function in reading comprehension [33]. It can therefore be assumed that reduced hippocampal volume and reduced prefrontal cortex thickness as features of people with low resilience levels may account for reading difficulties. Significantly, the resilience study did not show changes in the left parietal, temporal or occipital areas, which are involved in reading in most individuals [34]. Given that the act of reading is highly lateralized, it can be inferred that the efficiency of the reading process in individuals with varying levels of resilience may be reduced by functional and structural changes in areas related to stress control, emotionality and focus (that is, areas located in both hemispheres)—more so than those directly involving areas such as Broca’s or Wernicke’s [35].

There are still few studies available on the effects of resilience and carrier type on reading performance in healthy populations [36]. Access to such data could enable adults (including employers) to more consciously increase work performance by tailoring the information carrier to the individual’s predisposition related to resilience levels. 

The aim of the study was to investigate whether resilience differentiates reading comprehension scores in adults for informational texts presented on different media. Differences in the understanding of linear texts on different media have been the subject of research [11,37,38]. Most of them indicate in favor of texts on a paper medium. This is not dependent on one’s computer skills. However, no studies are available to test whether the level of cognitive resilience will differentiate the level of reading comprehension of people reading on different media.

Based on the available research, it was assumed that there would be differences in reading efficiency depending on the medium (i.e., people who read on paper will score better). It was also assumed that a high ability to cope with stress or adapt to new socio-economic and health conditions would result in better performance in this area regardless of the medium. 

In neurobiological terms, it was assumed that the overall neuronal mechanisms of reading on different media would be different and the level of resilience would modify these differences.

## 2. Materials and Methods

### 2.1. Research Group

The survey targeted adults aged between 20 and 69. The invitation and information about the survey were disseminated via email to adult employees of state institutions in the defined area. The inclusion criterion was the completion of an online application questionnaire (regarding epilepsy, brain injuries, neurological diseases, psychiatric diseases, taking permanent medication, dyslexia, visual defects). A total of 135 people (101 women and 34 men) were examined. The majority of the respondents were married (62.00%). Unmarried people accounted for 17.33% and those declaring an informal relationship for 10.66%. Only 6% of respondents were divorced and one was separated. One in three respondents were over 30 and under 40 years old. An almost equal group were respondents with an age range of 40–50 years (32.66%). More than one in five respondents were over 50 years old (22%), and almost one in ten were under 30 years old (8.67%). More than one in four respondents (28.67%) held a managerial position. Work seniority of more than half of the respondents (51.33%) was over 15 years. Only 4.66% of respondents had worked for more than 5 and less than 10 years. An equal group was made up of people whose length of service was between 6 months and 2 years. A period of employment of between 5 and 10 years characterized 14.67% of respondents. The remainder (21.33%) had been employed for more than 10 but less than 15 years.

### 2.2. Test Procedure

Results were obtained from three sources. The research procedure lasted 60 min. Before the start of the study, each person was asked to give written informed consent to participate in the study. They were also informed of the possibility of opting out of the study at any stage. From that moment, the data were coded and the code was used in further testing and neuroimaging of the brain.

Participants were asked to complete a questionnaire, which included information on demographics, and screening criteria forms to identify medical issues that may pose a risk of biased findings using NIRS. These data were used for exclusion–inclusion decisions, but not for hypothesis testing.

The research procedure began with a clause informing the potential respondent of the purpose, anonymity and voluntary participation in the study. The respondent was then given the Resilience Scale questionnaire—SPP-25—to complete [39]. The scale is designed for adults, both healthy and ill. It contains 25 statements on the various personality characteristics that make up mental resilience. Assessment is made on a 5-point Likert-type scale. In addition to the overall score, the scale measures the following five factors: Persistence and determination in action;Openness to new experiences and a sense of humor;Personal coping competence and tolerance of negative emotions;Tolerance of failure and treating life as a challenge;An optimistic attitude to life and an ability to motivate oneself in difficult situations [1].

High scores on the test indicated a high level of resilience, i.e., a high intensity of the traits mentioned above.

After completing the questionnaire, the subjects participated in a neuroimaging procedure using near-infrared spectroscopy. The protocol used is of author’s character. It is based on research on the hemodynamic activity of the brain during the reading of familiar words and information texts [40,41,42,43,44]. It included two tasks. During the first one, the subject was given a one-page informational text to read and later a single-choice test with 6 questions on the information content. The respondent was given a maximum of 3 min for the first task, and the same for the second. Each participant was randomly presented with an informative text and a test on paper or digital media [45].

The informational text was chosen for its neutrality and independence from religious views and socio-economic situation. Its content concerned information on planned training. It was single-page, so that the respondent would not be distracted by turning the page. The text met the requirements of an informational text by fulfilling the following:-objectivity—the facts are conveyed, and the author’s emotions and opinions are not uncovered;-reliability of sources—true information that does not lead the recipient into confusion or error;-written in a simple manner, understandable to any reader;-containing short, concise messages, providing complete information [46].

## 3. Results 

### 3.1. Analysis of Behavioral Data

Firstly, the results of the SPP-25 Scale questionnaire were analyzed. Based on the results, the study group was divided into three subgroups: those with low, average and high levels of resilience. The largest group of respondents showed high levels of resilience (42.22%), followed by medium (33.33%) and low (24.45%). In terms of responses to the reading efficiency test, respondents were able to give a maximum of six correct answers, with a minimum of zero. Most respondents gave four or five correct answers. A summary of the results is shown in Table 1.

To test whether there were significant differences in the number of correct answers between groups with different levels of resilience, the Kruskal–Wallis test for independent samples was used. The analysis showed no significant intergroup differences (df = 2; F = 0.791; *p* < 0.05). 

### 3.2. Analysis of Functional Near-Infrared Spectroscopy (fNIRS) Neuroimaging Data

NIRS is a neuroimaging technique that is increasingly being used to study cognitive function based on the non-invasive measurement of hemodynamic changes in the cortical brain surface, where activated brain areas experience high metabolic demand and increased oxygen uptake [47]. This oxygen consumption leads to an initial decrease in the amount of oxygenated hemoglobin (HbO), followed by an increase in regional cerebral blood flow, which consequently raises the HbO concentration [48]. NIRS uses the changing optical properties in cortical tissue by emitting near-infrared light into the cortex.

In the study, a 64-channel (32 × 32) fNIRScout system (NIRx Technology, Berlin, Germany) was used for the measurements, which consisted of 30 emitters and 22 detectors placed around the head according to the design shown in Figure 1. A total of 68 channels were obtained. Beta values were calculated in the bilateral prefrontal, temporal, parietal and occipital cortex at two wavelengths (750 and 850 nm). Raw fNIRS data were recorded using NIRStar14.2 acquisition software (NIRx, Germany) and then processed using Homer3 (version 1.52.0, NRSx Technology, Berlin, Germany) analysis software.

A total of 135 participants took part in the study. In the first phase, the results of left-handed individuals (due to potentially different lateralization of language areas), those with a history of neurological disease or brain injury and those taking medication that could affect cognitive function were eliminated from further analysis. The data of those whose records were incomplete (gaps in the record or missing markers)—a total of 41 participants—were also removed from further analysis.

The data obtained from the other respondents were pre-processed. To eliminate signal noise, the qt-nirs algorithm in the Matlab R2019b environment was used to remove channels with signal-to-noise ratios below 73% (SCI = 0.80; PSP = 0.10). Those that had the required signal-to-noise ratio in at least 20 individuals (54 channels) were left for statistical analyses. Based on these data, the final signal processing step eliminated those with less than 35% good channels. Ultimately, 73 people (48 women and 35 men) were selected for final statistical analyses. One in three respondents were over 30 and under 40 years old. An almost equal group were respondents with an age range of 40–50 years (30.98%). In contrast, more than one in four respondents were over 50 years old (28.17%), and only 5.63% were under 30 years old. Work seniority of more than half of the respondents (54.93%) was over 15 years.

Light intensities were then converted to optical densities and blood oxygen concentrations using modified Beer–Lambert laws. The signal was also bandpass-filtered from 0.01 to 0.5 Hz to remove both noise and interfering signals (heart rate, respiratory rate and Mayer waves). The initial time of the hemodynamic response function (HRF) was set to −2 s (i.e., baseline) and the end time to 60 s (i.e., the shortest time to complete the reading task and the test). Data prepared in this way were exported in .xlsx format to SPSS 28 software.

For the purpose of static analyses of the processed results, it was assumed that reading comprehension comprises two separate activities, i.e., reading the text and completing the test.

Analyses using the Kruskal–Wallis test (the distribution of results was not normal) showed that there were statistically significant differences in the neural activity of the brains of people with different levels of resilience in the two activities tested (see Figure 2).

Intergroup differences (at *p* = 0.05) for reading activities included channels 10 (F = 0.49) and 17 (F = 0.013), while for test completion they included channels 2 (F = 0.041), 38 (F = 0.031), 42 (F = 0.045), 43 (F = 0.026), 46 (F = 0.021), 57 (F = 0.050), 59 (F = 0.037), 61 (F = 0.042) and 62 (F = 0.015).

Intragroup analyses were then carried out to see whether the medium of information differentiated the number of correct answers. The Student’s *t*-test for independent groups was used for the analyses (see Figure 2). In relation to reading activity, the carrier differentiated neural activity:(a)in the high-resilience group: in channel 12 (t = 2.125; df = 27; *p* = 0.430);(b)in the medium-resilience group: in channels 61 (t = −2.133; df = 18; *p* = 0.47) and 68 (t = −2.472; df = 18; *p* = 0.24);(c)in the low-resilience group: in channels 2 (t = −2.51; df = 15; *p* = 0.24), 16 (t = −2.123; df = 15; t = 0.50) and 52 (t = −2.763; df = 15; *p* = 0.14).

In the task involving completing the test, the differences appeared (see Figure 3): (a)in the high-resilience group: in channels 54 (t = 2.087; df = 27; *p* = 0.023) and 56 (t = 2.193; df = 27; *p* = 0.019);(b)in the medium-resilience group: in channels 32 (t = −2.426; df = 18; *p* = 0.026), 40 (t = −2.962; df = 18; *p* = 0.004) and 50 (t = −2.187; df = 18; *p* = 0.042);(c)in the low-resilience group: in channel 56 (t = −2.775; df = 15; *p* = 0.014).

**Figure 3 brainsci-14-00944-f003:**
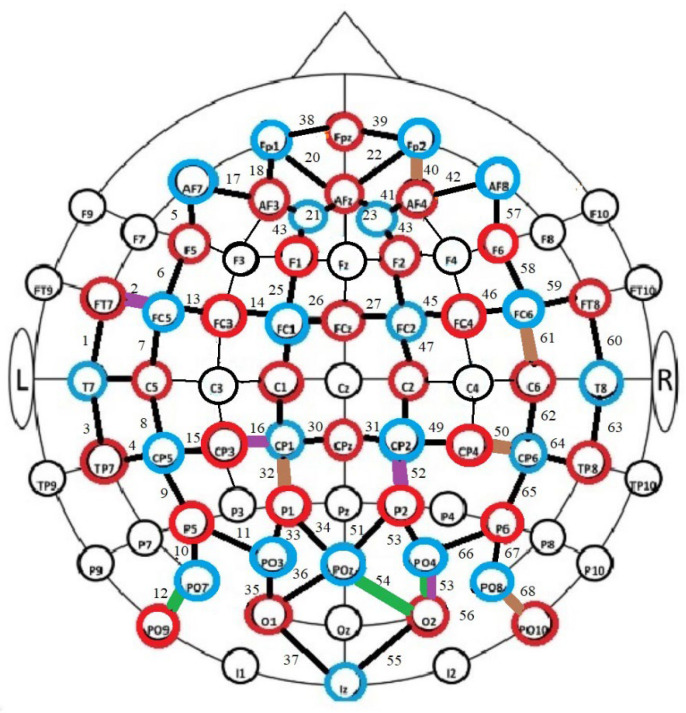
Graphical representation of significant intragroup differences in correct answers between those performing two tasks (reading and completing the test) on different media. Notes: green indicates channels where there were significant differences in the high-resilience group (both tasks); brown—in the medium-resilience group; purple—in the low-resilience group.

## 4. Discussion

Previous research has shown that the level of resilience differentiates cognitive performance among adults [49]. The results obtained do not support these conclusions. Regardless of the level of resilience in the groups studied, the number of correct answers was similar. This may be due to the nature of the text the respondents read. These results were also not differentiated by the medium of information. This is in line with a recent trend (which appeared only a few years ago), indicating that at the current level of technology development and literacy, the medium of information may not differentiate reading performance. For long texts, studies have shown that the type of medium had no effect on narrative comprehension, but in tasks on the chronology of events, reading on an electronic medium produced much worse results [47]. Similar results were obtained using a text that was a story of several pages, but for a short informational text the differences were insignificant [36]. These assumptions were confirmed by the results of the study presented here, in which a short informational text was chosen as neutral and frequently used in many professions. However, it only allows the examination of basic reading comprehension. It does not involve inference, comparison or summarization. On the other hand, informative texts often do not require a deep understanding of the content.

Differences occurring in one of the groups may be due to psychological conditions as a consequence of, for example, post-COVID-19 stress. Individuals with moderate levels of social and cognitive resilience are limited in their ability to adapt to changing conditions and to make use of available resources found in the environment. Among these individuals, skills in this area are further characterized by high variability and dependence on other factors. One of these may be the type of information medium. The Internet and technology during the epidemic provided an opportunity to connect with other people and thus maintain a sense of security, or an opportunity to release difficult emotions associated with isolation or the unpredictability of the situation. On the other hand, for a certain group of people, the imposed necessity to work remotely was a source of stress and difficulties related to the lack of time to prepare for the new situation and to improve their competences in the field [48]. Both these factors can affect stress levels, which in turn can determine cognitive activity. 

Notwithstanding the above, the level of resilience differentiated the neural mechanisms of the activities performed. Subject literature points to two supposed reasons for this: compensation and normalization. Researchers have suggested that increased activation during reading tasks in regions associated with general cognitive processing, including the right hemisphere and frontal and subcortical structures, reflects compensation for dysfunction of the reading system in the left hemisphere. The presumed compensatory processing can take various forms, such as increased reliance on working memory, attention, articulatory mechanisms and/or declarative memory to overcome reading difficulties, whereas normalization is usually inferred from increased activation in the “typical” reading network, which is thought to indicate the involvement of typical reading strategies through phonological decoding and/or rapid word recognition. A growing number of studies have begun to reveal intervention-related changes in grey matter volume, cortex thickness and white matter properties [36,50].

## 5. Conclusions

The aim of the study was to compare the reading comprehension performance of people with different levels of mental resilience. It was also tested whether the medium of information would further differentiate the results obtained and their neuronal mechanism. Reading comprehension testing involved two tasks: reading the text and completing the test.

Analyses showed no significant differences in the number of correct responses between groups with different levels of resilience. Within-group analyses showed no significant differences by medium in subjects with high and low levels of resilience. Such differences appeared in the group of people with moderate levels of resilience—those performing the tasks on paper medium performed better. Level of resilience was also a predictor of correct answers in this group of people.

Further analyses showed that the neural mechanisms in this scope were different for the study groups in both text reading and test completion activities. In the first task, the intergroup differences were related to areas of the left hemisphere, i.e., the inferior and medial frontal gyrus (Brodman area (BA) 46), the posterior temporal gyrus (BA 37) and the angular gyrus (BA 39). These areas are actively involved in the selective processing of linguistic material during the simultaneous presentation of multiple stimulus streams [47]. They are activated by both text and speech processing, particularly in terms of verbal fluency, lexical variety, sentence generation and word retrieval [51]. In terms of test completion, differences were found in both left and right hemisphere areas, i.e.:-the left prefrontal area (BA 10), responsible for understanding sentence syntax and inference while reading;-the left lower frontal gyrus (BA 44), involved in tasks requiring language fluency or sentence comprehension skills and lexical word variation;-the right prefrontal area (BA 10), which plays a key role in activities involving working memory, spatial memory and divided attention;-the right inferior frontal gyrus (BA 43,44,45,46), which—in addition to tasks similar to the area described above—is activated when the person is exposed to unintelligible speech and when the activity requires semantic and phonological processing, word generation, categorization or strategy formation [52,53].

The level of resilience therefore differentiated, in terms of both activities, between activity in frontal areas (involved in attentional and planning processes) and those directly involved in reading and speech [54]. 

Intragroup analyses showed that neuronal activity was differentiated by the type of information carrier. This means that in the group of people doing well with stress reduction and adaptation to new environments, reading comprehension using a paper medium resulted in greater neural activity in the left occipital and central areas. In the medium-resilience group, increased activation was evident in the digital media group and involved areas of the right hemisphere (frontal, temporal and occipital). In the last group, greater activity was noted in the left temporal and central parietal–occipital areas.

Near-infrared spectroscopy allows us to observe changes in the biochemical activities of specific areas of the brain. When an activity is automated, there is no noticeable inter- or intragroup difference. If, on the other hand, such a difference is evident, it may indicate over-activation and associated increased energy consumption (and consequently greater and faster task fatigue) or the involvement of additional areas in performing the given activity, resulting in a longer nerve impulse pathway and, consequently, longer task completion time [51]. 

Activation of occipital areas may indicate increased effort during activities involving the use of vision on different media. Activation of left parietal areas is associated with greater demands on visuospatial attention in tasks requiring episodic retrieval [51,52], whereas areas of the right hemisphere are involved in visual processing, concentration on spatial stimuli and the theory-of-mind mechanism. All these processes at each stage can be modulated by attention and emotion [55,56]. Groups of individuals with different levels of resilience, depending on the medium, presented different neural mechanisms that enabled them to successfully perform the reading comprehension task and achieve similar final results. It can therefore be concluded that the different neurobiological structure and functionality in individuals with different levels of resilience (due to genetic conditions or the number and quality of stressors) has its consequences in the neural mechanisms of the reading comprehension process.

Research has shown that in terms of reading short informational texts, the level of mental resilience does not differentiate reading performance. However, in the medium-level resilience group, a higher number of correct answers was found in those using a paper-based medium. This may indicate that this form of communication is more effective.

The results obtained in the studies discussed above also revealed the existence of other neural mechanisms of task performance—directly or indirectly related to the level of resilience. In practice, this means that depending on their level of resilience, adults may put different degrees of effort into a task. This can result in similar performance but, in the long term, increased fatigue and a faster burnout. To prevent this, it is worth implementing preventive measures as early as possible, tailored to the individual’s needs—and also to their level of mental resilience.

## 6. Limitations

The research presented here has several limitations. Reading speed was not analyzed and participants were not asked about their preferred reading medium. Moreover, with regard to informational texts, it would be worthwhile to investigate the efficiency of reading a text of several pages. This would make it possible to test reading comprehension at a deep level. 

## Figures and Tables

**Figure 1 brainsci-14-00944-f001:**
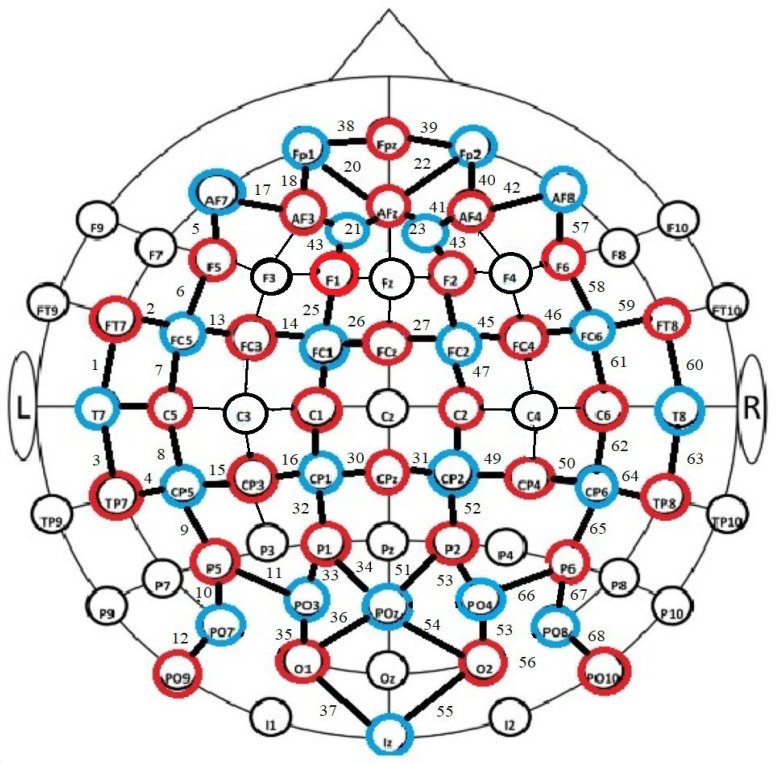
Schematic of optode placement in the study using fNIRS. Notes: red color indicates infrared light emitters, blue color indicates detectors; the numbers given are the numbers of the channels through which the data were read; Nz—nose, LPA—left hemisphere, RPA—right hemisphere.

**Figure 2 brainsci-14-00944-f002:**
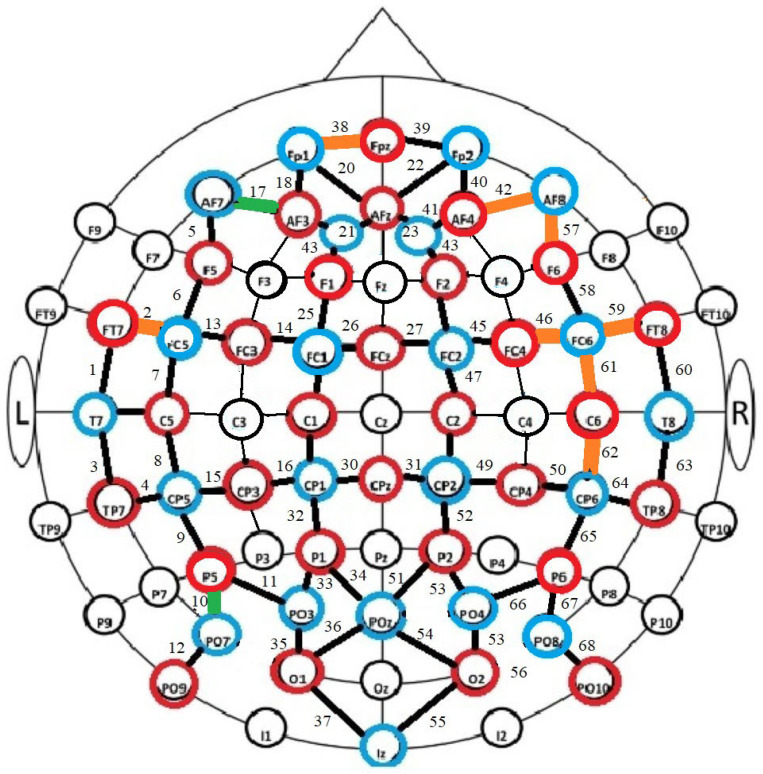
Graphical representation of significant differences in neuronal activity of groups with different levels of resilience. Notes: blue are detectors, red are emiters, green lines indicates areas with significant differences in text reading activities; orange—indicates differences in test completion activities.

**Table 1 brainsci-14-00944-t001:** Summary of results, regarding correct answers given by the respondents.

Number of Correct Answers	Number of Participants	Percentage of Participants among the Whole Sample
0	1	0.74
1	4	2.96
2	22	16.29
3	29	21.48
4	34	24.44
5	33	24.21
7	12	9.88

## Data Availability

Restrictions apply to the datasets, because the data is part of an ongoing study. Requests to access the datasets should be directed to the corresponding author.

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
