# Peer review of "Neuronal Mechanisms of Reading Informational Texts in People with Different Levels of Mental Resilience"

_brainsci, 2024, doi:10.3390/brainsci14090944_

Round 1
Reviewer 1 Report
Comments and Suggestions for Authors
I would like to refer to the article prepared as it is a good basis for the issue under study. I suggest the following changes and improvements to strengthen the text overall.
First, format the abstract according to journal standards and add keywords to comprehensively cover the subject under study and enhance indexing. Usually 5-7 words best describe the topic.
Better support the text of the introduction as there are points without bibliographic references.
At the end of the introduction, before your methodology, fill in the text of the purpose of the study, the literature gap essentially covered by your study or the innovation you propose as well as the individual objectives – sub-questions of your study.
In the methodology, mention the use of a questionnaire and that the sample was examined through it. I propose to paraphrase this point as the term used does not quite fit.
It is also important to make it clear how the recruitment was done, what audience you addressed, the duration of the study, the sampling and approval from bioethics – consent etc issues related to the study.
In the text there are various acronyms, eg NIRS, PTSD, please, for the sake of understanding by the readers, every first time a term is present in the text, it must be mentioned in full.
The protocol you followed, the procedure, the questionnaire used and the near-infrared spectroscopy procedure must be supported bibliographically by existing knowledge and other similar studies and in your theoretical background but also in the points to which reference is made - invocation in the methodology.
It is important to mention how long it took from the completion of the questionnaire to the processing of the near-infrared spectroscopy and whether your sample had any condition, comorbidity, or disease affecting hearing, vision, or reading.
It is also important to mention and support bibliographically why paper-based and digital media material was given for evaluation and to describe the content, linguistic, morphological, syntactic, complexity and grammatical content of the text.
The analysis of the data – your results should go under the results section and not within the methodology.
Similarly a part in 4. It is introductory and theoretical. This piece should be moved with the corresponding modifications to your theory while the results concerning the issue continue in the Results field.
The selection of 73 that were analyzed for MRI is of particular concern to me. This practice needs to be clinically and literature-supported as it increases bias if not properly supported as a common practice.
It is important to properly format the Discussion, which should be after the Results field (moved from this point to the current text format), and to add accordingly from the point now included in your Conclusions which is as shown part of your discussion as I see as I read it.
Also, the text and results must be formatted appropriately as the summary must be included in the Conclusions and the term Summary must be deleted.
Your conclusions must answer all the research questions posed and cover the purpose of the study.
Advisory, it would be good to develop the limitations of the study in terms of the structural points of the study and not only on issues that were not evaluated, to refer to these points in future extensions of your study, something that is missing in the present text.
Finally, write the abbreviations in the table where necessary or remove the footnote.
Reviewer 2 Report
Comments and Suggestions for Authors
- line 100: "or" should be substituted with "and". In fact, the prefrontal cortex is essential for information management and attention during a reading task, while the hippocampus contributes to comprehension through long-term memory.
- lines 105-109: reading is highly - but not exclusively - left lateralized. But stress and emotional control are differently distributed: e.g. negative emotions are rightly controlled, with a bilateral activation as well. The same holds for stress control. Focus control shows a bilateral activation.
- line 160: how did you define the subgroups and how the fitting of participants within them?
- lines 315-318: Here your overall assessment is that differences between participants with different levels of resilience impact on neural mechanism activation. How should this claim be related with the absence of differences in the obtained scores in the task? (at line 170 you state that there is no significant intergroup difference) For the best of my understanding, there are no differences in results between resilience groups, but only within one group depending on the medium.
- lines 363 and following: here you state that intergroup differences from the resilience level are in line with the literature. Sorry, at the very end it is not clear if there are not (see line 170) or there are (line 363).
Round 2
Reviewer 1 Report
Comments and Suggestions for Authors
Dear authors, you have addressed all my suggestions and improved the text.
Thank you
Author Response
Thank you for your response and for your time.
Reviewer 2 Report
Comments and Suggestions for Authors
From this second read the overall work appears a bit clearer. However, there are still unresolved problems.
The two fold analysis does not help, with the reading medium (between group analysis) and the resilience level (between group analysis) complicating the overall take-home message.
Overall, some grounding theoretical preliminary hypothesis are missing, and the discussion and interpretation of results appear to be post-hoc given.
In addition, results do not give a great support to the research by their own.
My suggestion is to be more focussed on one aspect only and to draw some preliminary hypotheses, and try to find a confirmation to them.
Author Response
Thank you for taking the time to re-examine the article.
Below are my suggestions, regarding the comments and suggestions you sent.
Overall, some preliminary theoretical hypotheses are missing, and the discussion and interpretation of the results seem to be presented post hoc.
Supplemented
Based on the available research, it was assumed that there would be differences in reading efficiency depending on the medium (i.e., those who read on paper would score better in this regard). It was also assumed that a high ability to cope with stress or adapt to new socioeconomic and health conditions would result in better performance regardless of the medium. On the neurobiological aspect, it was assumed that the overall neuronal mechanisms of reading on different carriers will be different and the level of resilience will modify these differences.
Moreover, the results obtained do not provide strong support for their research.
This type of research was carried out for the first time in a group of adults working only in clerical positions. Currently, this research is continuing with the participation of people whose work situation has changed even more during the epidemic.
My suggestion is to focus more on one aspect and formulate preliminary hypotheses and then try to find their confirmation.
As suggested, the hypotheses were supplemented. Data on within-group analyses have been removed.
Round 3
Reviewer 2 Report
Comments and Suggestions for Authors
I do appreciate the work done by the authors in this revision round.
However, there are still some issues that need further revision.
- line 107: Warnke' area? I suppose you wanted to mention Wernicke's brain area
- line 121: ... will score better in this area. Here "in this area" is not sufficiently clear, my suggestion is to remove it
- line 125: would modify these differences. I would use a smoother expression, as for example "would affect these differences"
- Table 1: here the caption should give more details. I suppose that "N" is the number of participants ho gave that given "Numbers of correct answers" and "%" is the percentage of the number of participants among the whole sample. If so, please help the reader to be sure about.
- lines 202-203: you claim that there is "no significant intergroup differences" but then you give a significant p-value p<0.05. How should it be read? Is the smaller-than sign supposed to be a "greater than"? Or is there a misunderstanding of the Krystal-Wallis test?
- line 226: insert a comma before "were eliminated .."
- lines 261-274: here you focus on differences in correct answers given by the medium (either screen or paper). However I cannot find reference to the medium, either from the text given in bullets or from the figure 3. Which one is paper and which is screen related result?
- lines 285-286: as a mere stylistic issue, I suggest you to change "a new trend" in a recent trend and erase "which only appeared a few years ago".
- line 296: the expression that "informative texts often do not require a deep understanding of the content" can hardly be accepted. I understand what you are referring to, since it is well clarified in the previous line (It does not involve inference, comparison or summarization". My suggestion is to modify "a deep understanding of the content" in "a deep beyond the literal text processing" .. or something like that. I agree with you that a text with some information does not require inference to the identification of eventual causative relations.
- lines 308-309: once again here I suggest you to be more smooth, and to change "determine the effectiveness of " in "may have affected their ..."
- Section 5: here you should be consistent. 46 Broadman area -BA should be Broadman area -BA 46, and 10 BA should be BA 10.
- lines 357-370: this section appears at the moment not perfectly clear, since there is no clear evidence of the medium /carrier differences (see a previous comment of mine on this topic (lines 261-274).
Author Response
Thank you very much for taking the time to review the article and your extremely important suggestions for making changes. The following are the responses and information, regarding the above.
As suggested, the text has been changed:
- line 107: Warnke's area? I suppose you meant to mention Wernicke's brain area
- line 121: ... will get a better score in this area. Here “in this area” is not clear enough, I suggest removing it
- line 125: would modify the differences. I would use a smoother expression, for example, “would affect these differences”
- Table 1: here the caption should provide more details. I suppose “N” is the number of participants who gave this “Number of correct answers” and “%” is the percentage of the number of participants in the whole sample. If so, help the reader to be sure.
- Line 226: insert a comma before “were eliminated....”
- lines 285-286: as a purely stylistic issue, I suggest changing the phrase “new trend” in recent trend and removing the phrase “which appeared only a few years ago”
- line 296: it is difficult to accept the expression that “informational texts often do not require a deep understanding of the content.” I understand what you are referring to, as it was well explained in the previous line (It does not include inference, comparison or summary). My suggestion is to modify “deep understanding of content” to “deep beyond literal text processing” ... or something like that. I agree with you that a text with some information does not require inference to identify possible cause-and-effect relationships.
- Lines 308-309: here again I suggest you be more fluid and change “determine the effectiveness” to “may affect their ...”
- Section 5: here you should be consistent. 46 Broadman -BA area should read Broadman -BA 46 area, and 10 BA should read BA 10.
Reviewer suggestion:
- lines 202-203: you state that there are no “significant between-group differences,” but then provide a significant p-value of p<0.05. How should this be read? Is the sign “less than” supposed to be “greater than”? Or is there a misunderstanding of the Krystal-Wallis test?
That is, with an assumed significance of p<0.05, the analyses showed no such differences
- lines 261-274: here you focus on differences in correct answers given by the medium (screen or paper). However, I can not find a reference to the medium, neither in the text given in the points, nor in Figure 3. Which is paper, and which is the result associated with the screen?
Supplemented the description of the figure to make it clearer
Figure 3. Graphical representation of significant intragroup differences in correct answers between those performing the Graphical representation of significant intragroup differences in correct answers between those performing two tasks (reading and completing the test) on different media. Notes: green indicates channels where there were significant differences in the high-resilience group (both tasks); brown - in the medium-resilience group; purple - in the low-resilience group.
Thank you again for your important comments.
